# Telemedicine in the Management of Patients with Rheumatic Disease during COVID-19 Pandemic: Incidence of Psychiatric Disorders and Fibromyalgia in Patients with Rheumatoid Arthritis and Psoriatic Arthritis

**DOI:** 10.3390/ijerph19063161

**Published:** 2022-03-08

**Authors:** Rosario Foti, Giorgio Amato, Ylenia Dal Bosco, Antonio Longo, Caterina Gagliano, Raffaele Falsaperla, Roberta Foti, Sergio Speranza, Francesco De Lucia, Elisa Visalli

**Affiliations:** 1Rheumatology Unit, Policlinico San Marco Hospital, 95123 Catania, Italy; rosfoti5@gmail.com (R.F.); giorgioamato@hotmail.it (G.A.); yleniadalbosco@gmail.com (Y.D.B.); robertafoti@hotmail.com (R.F.); spersergio@yahoo.it (S.S.); francescodelucia89@yahoo.it (F.D.L.); 2Eye Clinic, Policlinico San Marco Hospital, 95123 Catania, Italy; antlongo@unict.it (A.L.); caterina_gagliano@hotmail.com (C.G.); 3Pediatric Unit, Policlinico San Marco Hospital, 95123 Catania, Italy; r.falsaperla@ao-ve.it

**Keywords:** COVID 19, immune-rheumatic diseases, rheumatoid arthritis, psoriatic arthritis telemedicine

## Abstract

The management of patients with immuno-rheumatological diseases has profoundly changed during the COVID-19 pandemic and telemedicine has played an important role in the disease follow-up. In addition to monitoring disease activity and any adverse events, especially infectious events, assessing the psychological situation of the patient can be fundamental. Furthermore, COVID-19 has a serious impact on mental health and, since the beginning of the pandemic, a significantly higher incidence of anxiety disorders and depressive symptoms especially in younger people was observed. In this study, we evaluated the incidence of depressive disorders, anxiety, and fibromyalgia (FM) in our patients with rheumatoid arthritis and psoriatic arthritis during the lockdown period due to the COVID-19 pandemic and we validate the use of telemedicine in the clinical management of these patients. Mental and physical stress during the COVID-19 pandemic can greatly worsen FM symptoms and intensify patients’ suffering without a clinical flare of the inflammatory disease for patients affected by rheumatoid arthritis. Telemedicine has allowed us to identify patients who needed a face-to-face approach for therapeutic reevaluation even if not related to a flare of the inflammatory disease. Even if our data does not allow us to draw definitive conclusions regarding the effectiveness of telemedicine as greater than or equal to the standard face-to-face approach, we continue to work by modifying our approach to try to ensure the necessary care in compliance with safety and, optimistically, this tool will become an important part of rheumatic disease management.

## 1. Introduction

The management of patients with immuno-rheumatological diseases has profoundly changed during the COVID-19 pandemic [1] and telemedicine has played an important role in the disease follow-up [2].

Telemedicine is the remote delivery of healthcare services and clinical practices through medical data transmission via information and remote communication technologies (RCT). Recently, there has been an increase in publications about telemedicine and remote communication technologies (RCTs), therefore, an improved quality of available data can be documented [3]. It can represent an additional and potentially suitable tool for follow-up monitoring of patients, especially during the pandemic lockdown, and through it we were able to ensure specific treatment continuity for the management of inflammatory pathologies by identifying urgent remote situations, such as an infectious complication or a serious onset of the disease, that could require physical consultation [3,4].

In addition to monitoring disease activity and any adverse events, especially infectious events, assessing the psychological situation of the patient can be fundamental. Indeed, COVID-19 also has a serious impact on mental health, and Huang et al. demonstrated a significantly higher incidence of anxiety disorders and depressive symptoms, especially in younger people [5]. Depression and anxiety are frequently associated with fibromyalgia (FM) [6] which is one of the numerous comorbidities that may accompany inflammatory rheumatic diseases with possible interference with symptomatology, disease activity, and overall management plan [7].

The aim of this study is to evaluate the incidence of depressive disorders, anxiety, and fibromyalgia in our patients with rheumatoid arthritis and psoriatic arthritis during the lockdown period due to the COVID-19 pandemic and validate the use of telemedicine in the clinical management of these patients.

## 2. Materials and Methods

From 11 March 2020 to 11 May 2020 patients affected by rheumatoid arthritis and psoriatic arthritis treated with biological disease-modifying drugs afferent to the Rheumatology Unit of Policlinico S. Marco in Catania were contacted.

The synchronous telemedicine application is meant to offer a virtual alternative to the in-person Rheumatologist’s visit, and it requires a live interaction between health professionals and patients; this activity has been provided by telephone follow-up visits and by fax and e-mail usage in order to send reports to the patient. Technological improvements, combined with the high-speed internet and the massive spread of smartphones, enable the possibility to apply this framework and quickly deploy video teleconsultations from a patient’s home.

Patients were called to evaluate the state of health and the presence of any adverse events; laboratory test reports, such as acute phase reactants (erythrocyte sedimentation rate and C-reactive protein) and other tests to evaluate any liver and kidney dysfunction and blood counts were examined.

All patients with symptoms of infection temporarily withdrew biological disease-modifying anti-rheumatic drugs (bDMARD) or traditional disease-modifying anti-rheumatic drugs (tsDMARD) at the time of symptoms onset. A nurse administered the clinimetric questionnaires assessment to evaluate the disease activity, the impact of rheumatic disease on the health status, and the presence of anxiety, depression, and fibromyalgia.

More specifically, the following scales have been used.

The Rheumatoid Arthritis Impact of Disease (RAID), a patient-reported outcome measure evaluating the impact of rheumatoid arthritis (RA) on patient quality of life, has been used for patients with rheumatoid arthritis [8]. Moreover, the Psoriatic Arthritis Impact of Disease (PSAID) questionnaire has been used for patients with psoriatic arthritis (PsA): this questionnaire is made of 12 domains of health, each based on a 0–10 numerical rating scale (NRS) and each with a different weight on the final rating [9].

The presence of depressive symptoms (DS) has been assessed using the Beck Depression Inventory BDI-II, a 21-item self-report instrument that measures the severity of symptoms of depression, mild to severe, over the last two weeks with a threshold of 14; this instrument is used in several studies to examine the prevalence of DS [10,11].

The State-Trait Anxiety Inventory (STAI) questionnaire has been used to measure anxiety. It comprises separate self-report scales for measuring two distinct anxiety concepts: state anxiety and trait anxiety. The STAI-I scale consists of 20 statements that ask people to describe how they generally feel. The STAI-II scale also consists of 20 statements; however, the instructions require subjects to indicate how they feel at a particular moment in time. The STAI score ranges from a minimum of 20 to a maximum of 80. A low score indicates no or little anxiety while a higher score indicates a higher level of anxiety [12,13,14].

The Fibromyalgia Rapid Screening Tool questionnaire (FIRST) is a brief, self-administered questionnaire made of six “yes/no” questions to detect FM that has demonstrated high sensitivity and specificity among patients with chronic diffuse pain conditions and has been used in this study [15].

The Visual Analog Scale for Pain (VAS) for the assessment of pain [16] has been used for pain perception measurement.

Only patients who reported disease flare-ups or adverse events underwent an outpatient visit; in this case, composite clinimetric indices such as Disease Activity Score (DAS 28) [17] and Disease Activity Index for Psoriatic Arthritis (DAPSA) [18] were used.

In our clinical practice, the evaluation of fibromyalgia disorders and data related to this clinical assessment performed took place in the pre-pandemic period from 11 January 2020 to 11 February 2020, as well as the RAID and PSAID scores and composite clinimetric indices such as DAS 28 and DAPSA, which were analyzed.

## 3. Results

Overall, 171 patients affected by rheumatoid arthritis (RA) and 129 patients affected by psoriatic arthritis (PsA) were included in the study. Demographics and value at test are reported in Table 1. Between the two groups, no significant difference was seen in the VAS score (unpaired *t*-test *p* = 0.119).

FIRST (fibromyalgia indicator) was positive in 21.1% (RA) and 24% (PsA) of the patients (ns) and the rate increased in the two groups during the lockdown (RA *p* = 0.013, PsA *p* = 0.001), but no increase was observed between the two groups in the final rate.

No significant difference was seen in STAI (anxiety indicator) between the two groups; the rate increased in the two groups during the lockdown but no increase was observed between the two groups in the final rate.

The prevalence of depressive disorders (BDI positive) was higher in patients with RA than in those with PsA (respectively 24% and 13.2%, *p* = 0.026) with a mean value of BDI of 0.5 ± 0.9 in RA and of 0.2 ± 0.7 PsA (*p* = 0.006).

In the assessments performed before COVID-19, a significant correlation was found between impact of disease index and disease activity score (RA patients, RAID score and DAS 28 r = 0.572, *p* < 0.001; PsA patients, PSAID score, and DAPSA, r = 0.231, *p* = 0.008). No correlation between these parameters was detected in both groups in the assessment performed during lockdown, (RA patients (*n* = 50) r = 0.112, *p* = 0.438; PsA patients (*n* = 34) r = 0.131, *p* = 0.459) (Figure 1). 

Patients with RA with anxious symptoms (STAI positive) had a higher median value of RAID than patients who were STAI negative (STAI I: 5.00 vs. 0.57, STAI II 5.35 vs. 0.71, both Mann–Whitney *p* < 0.001). similarly, patients with PsA with anxious symptoms (STAI positive) had a higher median value of PSAID than patients who were STAI negative (STAI I: 3.25 vs. 2.85, Mann–Whitney *p* = 0.010; STAI II 3.6 vs. 2.4, Mann–Whitney *p* = 0.006).

Patients positive at FIRST (fibromyalgia indicator) had a higher median RAID score than those who were negative (median 4.93 vs. 1.40) (Mann–Whitney *p* < 0.001), while no difference was seen in the PSAID score (median 3.25 and 2.85, Mann–Whitney *p* = 0.969).

Patients positive at BDI had a higher RAID score (median 6.28 vs. 1.14, Mann–Whitney *p* < 0.001) and PSAID score (median 4.95 vs. 2.85, Mann–Whitney *p* = 0.003) than those negative at both parameters.

In 50 RA patients (mean age 61 years) that were reexamined, STAI I and STAI II were positive in 42 (median score respectively of 51.5 and 46); median VAS was 70, mean RAID was 6.28; median FIRST pre was 0, median DAS 28 pre-COVID-19 was 4.3, and during COVID-19 was 4.95. No correlation was seen between the DAS 28 score and the RAID score during COVID-19.

In 34 PsA patients (mean age 59 years) that were reexamined, STAI I was positive in 27 (median score 50) and STAI II was positive in 28 (median score 47); median PSAIS was 3125; median FIRST pre was 8, DAPSA pre-COVID-19 was 12, and DAPSA during COVID was 8.

No correlation was seen between the PSAID score and DAPSA score during COVID-19.

## 4. Discussion

As a Rheumatology Department, we have switched approximately 80% of outpatient appointments to synchronous telemedicine. This has worked surprisingly well and patients have been very understanding. Outpatient clinic face-to-face consultations are limited to urgent patients. Patient management through telemedicine has allowed us to carry out a remote assessment of the state of health and of the psychological implications that the changes related to the COVID-19 pandemic have determined in our patients without exposing them to an increased infectious risk.

In the current study, we found a high incidence of fibromyalgia and high levels of subjective perception of disease worsening in patients with rheumatoid arthritis and psoriatic arthritis during the COVID-19 outbreak and lockdown measures.

Physical and psychological trauma and exposure to chronic stress directly and negatively affect the underlying process of central sensitization and they are well-known factors associated with triggering and exacerbating chronic pain and FM symptoms [19,20]. Therefore, it is probable that traumatic or very stressful life events are not the actual cause of FM, but in genetically predisposed individuals these factors may contribute to the mismodulation of brain circuitries involved in pain and emotional processing [21], constituting the link between significant psychological factors and FM symptoms.

The COVID-19 pandemic represents a unique stressful condition that causes radical personal, family, professional, economic, and social changes, therefore the restraints imposed by the lockdown had a variable impact on the well-being of patients with rheumatic diseases; in fact, in some people this period disrupted a delicate physical and psychosocial balance and resulted in a worsening of disease severity, whereas for others this time brought about the opportunity to introduce beneficial changes in daily and working habits, which ultimately resulted in improvements in well-being [19,20,22].

Several studies have reported mental health problems among the general population during the COVID-19 pandemic and especially in the early phase of the COVID-19 outbreak, a range of psychiatric morbidities, including persistent depression and anxiety were reported [23,24].

Rheumatic diseases such as rheumatoid arthritis [25] and psoriatic arthritis [26] are known to have a high association with psychiatric conditions such as depression and anxiety and the COVID-19 outbreak results in greater stress with consequent worsening of pre-existent mental health disorders or the ensuing of new ones [24].

Bathia et al. suggest that patients with arthritis in general and rheumatoid especially may be at additional risk for worsening mental health during the pandemic while evidence on the direct and indirect impact of the situational and the disease-related factors is not yet known [27].

Although the evaluation of depression and psycho-affective aspects, in general, requires complex tools and questionnaires [28,29], our data obtained with simple tests, BDI and STAI I and II, showed that the group of patients with rheumatoid arthritis report significantly higher levels of depression, while there were no significant differences in the two groups regarding the anxious state.

Discordance between outcomes assessed by composite measures based on inflammation versus patient-reported outcomes can often be found in the management of RA and PsA, however, it is crucial to include both aspects in this context. The RAID and PSAID scores have been used during the COVID-19 outbreak for disease follow-up; 50 patients with RA and 34 with PsA with high disease activity evaluated through these questionnaires underwent an outpatient visit.

For PsA patients, no correlation was found between the DAPSA and PSAID scores; likewise, for RA patients no correlation was found between the DAS28 and RAID scores. For these patients, clinical and laboratory data did not ultimately match with the feeling of the worsening of the disease reported by questionnaires obtained with the use of telemedicine, however, there was an increase in fibromyalgia symptoms and a correlation with the levels of anxiety and depression.

The RAID and PSAID scores work well as patient-reported outcomes in routine care and we have found these questionnaires to be simple and easy to incorporate into the routine care setting for patients with RA and PsA.

Mistry et al. demonstrated that patients with RAID < 2 are also in the DAS28 categories of low disease activity (LDAS) or DAS28 remission (RDAS). Furthermore, it provides a potential time-saving utility by avoiding face-to-face disease activity assessments for these patients. RAID reveals a high burden of unmet needs in patients in RDAS/LDAS with scoring ≥ 2 and the scrutiny of the seven domains assessed provides individualized opportunities for improved RA management, especially for fatigue and sleep problems [30].

Likewise, in PsA, a discrepancy between patients’ and physicians’ ratings of general health status was demonstrated and factors associated with discordance were psychological rather than physical and these were more frequent in patients in remission. On the other hand, the PSAID questionnaire surely captures many patients’ perceptions that are not necessarily related to disease activity. For truly holistic care, there should be two treat-to-target goals, one based on an inflammation-derived measure and one on a patient-reported outcome [31].

## 5. Conclusions

In conclusion, mental and physical stress during the COVID-19 pandemic can greatly worsen FM symptoms and intensify patients’ suffering without a clinical flare of the inflammatory disease for patients affected by rheumatoid arthritis.

Telemedicine has allowed us to identify patients who require a face-to-face approach for therapeutic reevaluation even if not related to a flare of the disease.

Even if our data does not allow us to draw definitive conclusions regarding the effectiveness of telemedicine as greater than or equal to the standard face-to-face approach, we continue to work by modifying our approach to try to ensure the necessary care in compliance with safety and, optimistically, this tool will become an important part of rheumatic disease management.

## Figures and Tables

**Figure 1 ijerph-19-03161-f001:**
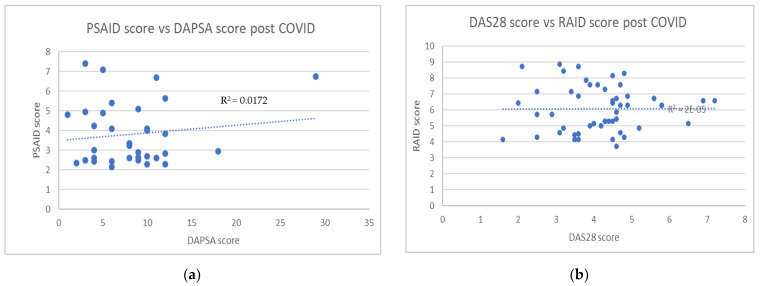
In RA patients (*n* = 50), no correlation was found between DAS28 and RAID score (r = 0.112, *p* = 0.438) (**a**) when comparing values detected before and after the COVID-19 pandemic; likewise, in PsA patients (*n* = 34), and no correlation was found between DAPSA and PSAID score (r = 0.131, *p* = 0.459) (**b**).

**Table 1 ijerph-19-03161-t001:** Demographics and value at test.

	Rheumatoid Arthritis (171)	Psoriatic Arthritis (129)	*p*
Age	57.0 ± 11.1	56.2 ± 10.1	0.110
VAS	33.0 ± 31.3	27.8 ± 26.5	0.119
BDI	41 (24%)	17 (13.2%)	0.026
BDI score	0.5 ± 0.9	0.2 ± 0.7	0.006
STAI I	64 (37.4%)	36 (27.9%)	0.108
STAI II	56 (32.7%)	38 (29.5%)	0.615
STAI I score	38.5 ± 13.6	35.4 ± 11.3	0.030
STAI II score	36.0 ± 11.9	33.9 ± 11.0	0.118
FIRST	36 (21.1%)	31 (24%)	0.576

## Data Availability

Not applicable.

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
