# Peer review of "Telemedicine in the Management of Patients with Rheumatic Disease during COVID-19 Pandemic: Incidence of Psychiatric Disorders and Fibromyalgia in Patients with Rheumatoid Arthritis and Psoriatic Arthritis"

_ijerph, 2022, doi:10.3390/ijerph19063161_

Round 1
Reviewer 1 Report
This paper proposes a telemedicine in the management of patients with rheumatic disease during Covid-19 pandemic. Concretely, the incidence of psychiatric disorders (depression and anxiety) and fibromyalgia in patients with rheumatoid arthritis and psoriatic arthritis is evaluated. Moreover, a telemedicine in the management of these patients using indexes (scales) that can be corrected by the clinimetric questionnaires is proposed.
This paper is important for the clinical management of patients with rheumatic disease during Covid-19 pandemic. However, there are some insufficient explanations, that should be added and/or corrected, as shown in the followings.
(1) About the evaluation of patients with rheumatic disease using RAID and PSAID (3. Results)
There are no correlations between scores of RAID, PSAID using in the telemedicine and those of DAS28, DASPSA using in the face-to-face medicine after detected Covid-19 pandemic. If this is true, are RAID and PSAID considered reasonable for evaluating patients with rheumatoid arthritis and psoriatic arthritis? If so, the reason should be clearly described.
(2) About the clinical management using telemedicine (2. Materials and Methods)
・In section 2 (Materials and Methods), the clinical management for patients with rheumatic disease is difficult to understand, because the telemedicine and the face-to-face medicine are described together. Indexes (scores) of telemedicine and those of face-to-face approach should be shown separately.
・Moreover, the conditions separating the telemedicine and the face-to-face approach should be described, if possible.
(3) The following expressions, such as abbreviations, should be reconsidered.
・For example, if rheumatoid arthritis is expressed as RA, psoriatic arthritis as PsA, in line111- line136, AR→RA and AP→PsA.
・line127- line136 and Table 1, STAIâ… → STAI-S ?, STAIâ…¡→ STAI-T ?
・line18; fibromyalgia → fibromyalgia (FM)
・line38; RCTs → remote communication technologies(RCTs) ?
etc.
Author Response
Dear Editor and Reviewers
In our review we have revised english language and style and clarified research design.
In particular we have added and corrected as follows.
(1) About the evaluation of patients with rheumatic disease using RAID and PSAID (3. Results) There are no correlations between scores of RAID, PSAID using in the telemedicine and those of DAS28, DASPSA using in the face-to-face medicine after detected Covid-19 pandemic. If this is true, are RAID and PSAID considered reasonable for evaluating patients with rheumatoid arthritis and psoriatic arthritis? If so, the reason should be clearly described.
Regard this consideration, we reviewed the role in clinical practice regarding use of RAID and PSAID and supporting our result with literature that has been included in discussion.
(2) About the clinical management using telemedicine (2. Materials and Methods)
・In section 2 (Materials and Methods), the clinical management for patients with rheumatic disease is difficult to understand, because the telemedicine and the face-to-face medicine are described together. Indexes (scores) of telemedicine and those of face-to-face approach should be shown separately.・
In section 2 (Material and Method) we have described the clinical management for patients with rheumatic disease during pandemic period which provided for use of telemedicine and in particular the use of questionnaires described. Face to face medicine was used only for patients who had disease flare assessed by telemedicine. During the visit were used composite clinimetric indices such as DAS 28 e DAPSA.
Moreover, the conditions separating the telemedicine and the face-to-face approach should be described, if possible.
Face to face medicine was used only for patients who had disease flare assessed by telemedicine and face to face approach has been described
(3) The following expressions, such as abbreviations, should be reconsidered.
・For example, if rheumatoid arthritis is expressed as RA, psoriatic arthritis as PsA, in line111- line136, AR→RA and AP→PsA.
・line127- line136 and Table 1, STAIâ… → STAI-S ?, STAIâ…¡→ STAI-T ? This data has been corrected
・line18; fibromyalgia → fibromyalgia (FM) This data has been corrected
・line38; RCTs → remote communication technologies(RCTs) ? This data has been corrected
etc.
the manuscript was reviewed and corrected

Reviewer 2 Report
improve and revise english language in the abstract and introduction sections (lines 16,17, 37,38).
line 72 tsDMARDS ( transitional synthetic disease modifying agents).
is there an open name for the VAS pain scale ( line 102).
results section:
please revise RA and PsA abbreviations, (AR and AP has been used instead in some places in the results section, line 111, 114 and others, unless you are referring to something else).
capitalize (first) in line 111
please report differences in numerical and statistical significance . it is not very clear from the results to the conclusion section whether the comparison was made between psA and RA patients or each group compared to the pre-pandemic stats? make sure to clarify.
results section, second paragraph line 113, what was the pre-pandemic numbers in jan and feb? can you specify the date range, can you extend to 3 months since the pandemic results were based on 3 month duration (march-may).)
please specify the duration in which the pre pandemic items were investigated?
discussion:
line 177: rheumatoid arthritis
line 183-185, here the comparison is between RA and PsA rather than RA pre and during pandemic and PsA pre and during pandemic.
---
overall great idea,
in table 1, please provide the pre-pandemic results as well, and clarify the duration in which those test were done.
please improve the reporting of the results section and reflect in the discussion and conclusion.
Author Response
Dear Editor and Reviewers
In our review we have revised english language and style and clarified research design.
In particular we have added and corrected as follows.
- Improve and revise english language in the abstract and introduction sections (lines 16,17, 37,38); This data has been corrected
line 72 tsDMARDS ( transitional synthetic disease modifying agents). This data has been corrected
is there an open name for the VAS pain scale ( line 102). This data has been corrected
Manuscript has been reviewed and corrected
- Results section:
please revise RA and PsA abbreviations, (AR and AP has been used instead in some places in the results section, line 111, 114 and others, unless you are referring to something else). This data has been corrected
Capitalize (first) in line 111 this data has been corrected
please report differences in numerical and statistical significance . it is not very clear from the results to the conclusion section whether the comparison was made between psA and RA patients or each group compared to the pre-pandemic stats? make sure to clarify.
pre pandemic period in which items were investigated was January 11th 2020 to February 11th 2020 and items during pandemic was investigated from March 11th 2020 to May 11th 2020.
During this period were collected data on Fibromyalgia, RAID, PSAID, DAS 28 e DAPSA with comparison between Fibromyalgia Pre e during pandemia, RAID and DS 28 pre e during pandemia and PSAID and DAPSA pre e during pandemia. The results section has been modified.
results section, second paragraph line 113, what was the pre-pandemic numbers in jan and feb? can you specify the date range, can you extend to 3 months since the pandemic results were based on 3 month duration (march-may).)
pre pandemic period in which items were investigated was January 11th 2020 to February 11th 2020, items during pandemic was investigated from March 11th 2020 to May 11th 2020.
please specify the duration in which the pre pandemic items were investigated? Pre Pandemic items were investigated from January to February 2020.
- discussion: line 177: rheumatoid arthritis
line 183-185, here the comparison is between RA and PsA rather than RA pre and during pandemic and PsA pre and during the pandemic.
Here the comparison is between RA and PsA during the pandemic, no data on anxiety and depression are available in the pre-pandemic period.
overall great idea,
in table 1, please provide the pre-pandemic results as well, and clarify the duration in which those tests were done.
In table 1 only data relating to telemedicine during the pandemic period were reported ( FIRST, STAI and BDI). In the pre-pandemic period were collected data on Fibromyalgia, RAID, PSAID, DAS 28 e DAPSA with the comparison between Fibromyalgia Pre e during pandemic ( values reported in the results). Comparison between RAID and DAS 28 during pandemic and PSAID and DAPSA during pandemic were reported in figure 1. Comparison between RAID and DAS 28 pre-pandemic and PSAID and DAPSA during pandemic were reported following (Can another figure be included in the manuscript?).
In AR patients, a significant correlation was found between RAID and DAS 28 (r=0.572, p<0.001)
In PA, patients, a significant correlation was found between PSAID and DAPSA (r=0.231, p=0.008)
please improve the reporting of the results section and reflect in the discussion and conclusion.
The manuscript has been reviewed and improved

Round 2
Reviewer 1 Report
This paper proposes a telemedicine in the management of patients with rheumatic disease during Covid-19 pandemic. The incidence of psychiatric disorders (depression and anxiety) and fibromyalgia in patients with rheumatoid arthritis and psoriatic arthritis is evaluated. Then, a telemedicine in the management of these patients using indexes (RAID, PSAID, etc.) that can be corrected by the clinimertic questionnaires is proposed.
The proposed telemedicine seems to be useful especially during Covid-19 pandemic, and this paper is acceptable. However, it is advised to correct the careless mistakes such as the followings.
For example,
・line 83 and line 230 ; PsAID → PSAID
・line 110 ; as DAS28 and (DAPSA) → as DAS28 and DAPSA
・line 152 ; PSAI score → PSAID score
・line 158 ; in in PsA patients → in PsA patients
Author Response
Dear Editor and Reviewer
Thank you for your comments and suggestions. We have revised and corrected the reported careless mistakes and in particular:
line 83 and line 230 ; PsAID → PSAID
line 110 ; as DAS28 and (DAPSA) → as DAS28 and DAPSA
line 152 ; PSAI score → PSAID score
line 158 ; in in PsA patients → in PsA patients
Kinds regards
Dott ssa Visalli Elisa